# REGRET-OPTIMAL LIST REPLICABLE BANDIT LEARNING: MATCHING UPPER AND LOWER BOUNDS

**Michael Chen**
Iowa State University
mqychen@iastate.edu

**A. Pavan**
Iowa State University
pavan@iastate.edu

**N. V. Vinodchandran**
University of Nebraska - Lincoln
vinod@unl.edu

**Ruosong Wang**
Peking University
ruosongwang@pku.edu.cn

**Lin F. Yang**
University of California, Los Angeles
linyang@ee.ucla.edu

## ABSTRACT

This paper investigates *list replicability* Dixon et al. (2023) in the context of multi-armed (also linear) bandits (MAB). We define an algorithm $A$ for MAB to be $(\ell, \delta)$-list replicable if with probability at least $1 - \delta$, $A$ has at most $\ell$ traces in independent executions even with different random bits, where a trace means sequence of arms played during an execution. For $k$-armed bandits, although the total number of traces can be $\Omega(k^T)$ for a time horizon $T$, we present several surprising upper bounds that either independent of or logarithmic of $T$: (1) a $(2^k, \delta)$-list replicable algorithm with near-optimal regret, $\widetilde{O}(\sqrt{kT})$[1], (2) a $(O(k/\delta), \delta)$-list replicable algorithm with regret $\widetilde{O}\left(\frac{k}{\delta}\sqrt{kT}\right)$, (3) a $((k+1)^{B-1}, \delta)$-list replicable algorithm with regret $\widetilde{O}(k^{\frac{3}{2}}T^{\frac{1}{2}+2^{-(B+1)}})$ for any integer $B > 1$. On the other hand, for the sublinear regret regime, we establish a matching lower bound on the list complexity (parameter $\ell$). We prove that there is no $(k - 1, \delta)$-list replicable algorithm with $o(T)$-regret. This is optimal in list complexity in the sub-linear regret regime as there is a $(k, 0)$-list replicable algorithm with $O(T^{2/3})$-regret. We further show that for linear bandits with $d$-dimensional features, there is a $\widetilde{O}(d^2 T^{1/2+2^{-(B+1)}})$-regret algorithm with $((2d + 1)^{B-1}, \delta)$-list replicability, for $B > 1$, even when the number of possible arms can be infinite.

## 1 INTRODUCTION

The issue of replicability and reproducibility in science has become a significant concern, attracting substantial attention from the broader scientific community Begley & Ellis (2012); Ioannidis (2005); Baker (2016); of Sciences et al. (2019). Recently the machine learning community also started addressing this issue. Inspired by reproducibility workshops at ICML/ICLR, the ML community has created a dedicated online conference, the ML Reproducibility Challenge (MLRC) Sinha et al. (2023), to encourage the publishing and sharing of scientific results that are reliable and reproducible. In machine learning, a common approach to ensure replicability and reproducibility is to make datasets and code publicly available. However, it is unclear if such an approach is sufficient, as machine learning algorithms draw samples from data distributions and are usually randomized. This inherent randomness results in non-replicability. Therefore, it is more desirable to design machine learning algorithms that are replicable, i.e., algorithms that return the same result over multiple runs, even when different runs observe different sets of samples from the data distribution. The aforementioned requirement has recently led to theoretical studies and rigorous definitions of replicability Impagliazzo et al. (2022); Dixon et al. (2023), with various notions of replicability being proposed.

In this work, we study bandit algorithms with replicability guarantees. The Multi-Armed Bandit (MAB) is a learning task in which a learner interacts with an environment consisting of a set of fixed $k$ distributions (called arms) in discrete time steps. In each of the $T$ steps, the learner selects

---

[1]Here $\widetilde{O}(\cdot)$ omits $\log k$, $\log \log T$, $\log \frac{1}{\delta}$.

an arm and receives an i.i.d. sample from that arm, called the reward. The goal of the learner is to maximize the cumulative expected reward, equivalently minimizing the "regret" (defined formally later), which characterizes the total sub-optimality of the actions played over the learning course.

Ideally, an MAB algorithm with *perfect replicability* should always select exactly the same trace (i.e., sequence of arms) over multiple runs, even if the algorithm receives a different set of samples from the data distribution for each run. However, as shown later in the paper (and discussed in Dixon et al. (2023)), it is impossible for MAB algorithms with perfect replicability guarantees to have a non-trivial regret bound, as algorithms need to adapt to the randomness from the history to learn effectively. Therefore, we ask, *what is the strongest notion of replicability of a bandit algorithm without suffering significant regret loss?*

Inspired by a recent study Dixon et al. (2023) that investigates replicability in the supervised learning setting, we propose a notion of *list replicability* for bandit learning. In this setting, we require that the trace (sequence of actions) produced by the algorithm during the entire learning course be predictable to a certain extent. That is, no matter how many times we repeat the algorithm for the bandit instance, the resulting sequence of actions comes from a small finite list. Note that in general, a $T$ horizon MAB algorithm with $k$ arms can have $k^T$ possible traces. Such a guarantee would be useful for bandit learning in safety-critical applications, as one can be prepared for the action sequence being played. Bandit algorithms and more general reinforcement learning algorithms are increasingly used in clinical trials and social experiments to personalize treatment decisions for individuals Zhang et al. (2024). In such domains, it is highly desirable to limit the possible sequence of actions and limit uncertainty.

The formalization of the concept of replicability in various computational settings is a significant and emerging area of research. This work takes an important step toward understanding the foundational principles of replicability in the context of bandit problems.

## 2 PRELIMINARIES

For $n \in \mathbb{N}$, we use $[n]$ to denote the set $\{1, \ldots, n\}$ and for $a, b \in \mathbb{R}$ we use $[a, b]$, $(a, b)$, $[a, b)$, and $(a, b]$ to represent the closed, open intervals, and their corresponding half-open intervals respectively. We use $\tilde{O}(\cdot)$ to subsume $\log k$, $\log \log T$, and $\log \frac{1}{\delta}$. In this paper, we consider the following Multi-Armed Bandit (MAB) instance.

**Assumption 2.1.** *An MAB instance is a tuple $(D, T)$ where $D = (D_1, \ldots, D_k)$ is a collection of distributions and $T \in \mathbb{N}$ is the horizon such that[2] each $D_i$ is supported on $[0, 1]$ and has a mean $\mu_i$. For any $t \in [T]$, if action $a_t \in [k]$ is played, the reward $r_t$ is an i.i.d. sample drawn from $D_{a_t}$.*

Here, we use $k$ to denote the number of arms and $T$ to denote the horizon. For a collection of means $\mu_1, \ldots, \mu_k$ we use $\mu^* = \max_{a \in [k]} \{\mu_a\}$ to denote the highest mean and for each $i \in [k]$, $\Delta_i = \mu^* - \mu_i$ to denote its reward gap. An arm $i$ with $\Delta_i = 0$ is called an optimal arm; otherwise, it is called a sub-optimal arm. Let $\mathcal{M}$ be an algorithm for MAB. The algorithm $\mathcal{M}$ defines probability distribution $P_{\mathcal{M}}$ on traces $[k]^T$ in the following sense: Given a trace $a \in [k]^T$, probability of $a$, denoted $P_{\mathcal{M}}(a)$, is the probability that $\mathcal{M}$ takes trace $a$. For a set $S$ of traces, $P_{\mathcal{M}}(S) = \sum_{a \in S} P_{\mathcal{M}}(a)$.

**Definition 2.2** ($(\ell, \delta)$-MAB Replicability). *An algorithm $\mathcal{M}$ for MAB is $(\ell, \delta)$-list replicable if for every MAB instance $(D, T)$, there exists a set of traces $S_{D,T} \subseteq [k]^T$ such that $|S_{D,T}| \leq \ell$ and $P_{\mathcal{M}}(S_{D,T}) \geq 1 - \delta$.*

*When $\delta = 0$, then we just say that $\mathcal{M}$ is $\ell$-list replicable. We call $\ell$ the list complexity of $\mathcal{M}$.*

**Definition 2.3** (Regret). *We measure the performance of an algorithm by the* regret, *which is $R_T = \sum_{t=1}^{T} \max_a \mu_a - \mu_{a^t}$, where $a^t$ is the action played at time $t$. An MAB algorithm is effective if the regret is sublinear in $T$: in this case, the average regret per round is approaching $0$ as $T \to \infty$.*

Trivially, any MAB algorithm is $k^T$-list replicable because a trace must be an element of $[k]^T$; and an algorithm playing the same action all the time is a $1$-list replicable algorithm, yet with regret $T$. We aim to create MAB algorithms that have both low regret and are $(\ell, \delta)$-list replicable with small $\ell$ and $\delta$.

---

[2]For presentation simplicity, we restrict to the $[0, 1]$ setting. Our results generalize to sub-Gaussian distributions straightforwardly.

For the linear bandits, we assume that the rewards $r_t$ satisfy a linear structure. Specifically,

**Assumption 2.4.** *For any $a \in \mathcal{A}$, we assume $\|a\|_2 \leq 1$ and hence $\mathcal{A}$ is a compact set. For any time $t$, if action $a_t$ is played, the reward $r_t$ satisfies $r_t = \langle a_t, \theta^* \rangle + \eta_t$ where $\theta^*$ is unknown with $\|\theta^*\|_2 \leq \sqrt{d}$, and $\eta_t$ is an history independent mean-$0$ $\sigma$-sub-Gaussian random variable that satisfies $\mathbb{E}[\eta_t] = 0$ and $(\forall x > 0). \Pr[|\eta_t| > x] \leq 2\exp\left(\frac{-x^2}{\sigma^2}\right)$.*

We denote $\mu_a = \langle a, \theta^* \rangle \in [0, 1]$ as the unknown true mean of an arm $a$, and let $a^* = \arg\max_{a \in \mathcal{A}} \langle a, \theta^* \rangle$, which is the unknown optimal action. For presentation simplicity, we assume $\sigma = 1$.

## 3 OUR CONTRIBUTIONS

In this work, we introduce a viable notion of replicability which is called *list replicability* in the context of Multi-Armed Bandit algorithms. We summarize our technical results below.

In Section 5.1, we present a $(2^k, \delta)$-list replicable MAB algorithm with near-optimal regret, $\tilde{O}(\sqrt{kT})$. Importantly, the total number of possible traces is completely independent of $T$.

In Section 5.2, we present a $(O(k/\delta), \delta)$-list replicable MAB algorithm with $\tilde{O}(k/\delta\sqrt{kT})$ regret. When $\delta$ is constant, the total number of possible traces is $O(k)$, while the regret bound is $\tilde{O}(\sqrt{\text{poly}(k)T})$.

In Section 5.3, we present a $((k + 1)^{B-1}, \delta)$-list replicable algorithm with regret bound $\tilde{O}(k^{\frac{3}{2}}T^{\frac{1}{2}+2^{-(B+1)}})$ for any $B > 1$. By setting $B$ to be a constant, the total number of possible traces is polynomial in $k$, while the regret bound is sublinear in $k$. By setting $B = \log\log T$, the total number of possible traces is $k^{O(\log\log T)}$, while the regret bound is $\tilde{O}(k^{3/2}\sqrt{T})$.

In Section 6, we further show that the guarantees of our third algorithm are nearly tight by establishing a $(k-1)$ lower bound for any list replicable algorithm with failure probability at most $1/(k+1)$ and regret $o(T)$, almost exactly matching the $k$-list replicable upper bound for $B = 2$. Interstingly, we use a cubical version of Sperner/KKM lemma for establishing the lower bound.

Finally, in Section 7, we generalize our third algorithm to the linear bandit setting. For a $d$-dimensional linear bandit instance, we present a $((2d + 1)^{B-1}, \delta)$-list replicable algorithm with regret $\tilde{O}(d^2T^{\frac{1}{2}+2^{-(B+1)}})$ for any $B > 1$.

## 4 RELATED WORK

Bandit algorithms have been extensively studied in the general setting Lattimore & Szepesvári (2020); Slivkins et al. (2019); Bubeck et al. (2012); Auer et al. (2002); Cesa-Bianchi & Fischer (1998); Kaufmann et al. (2012). In the MAB setting, Thomson sampling and UCB achieve an optimal regret of $\tilde{O}(\sqrt{kT})$ Agrawal & Goyal (2017); Auer et al. (2002). In the $d$-dimensional linear bandit setting, Contextual Thomson sampling and LinUCB achieve a regret of $\tilde{O}(d\sqrt{T})$ Lattimore & Szepesvári (2020). There is extensive work in other bandit variants, and interested readers are referred to Lattimore & Szepesvári (2020).

Several works Impagliazzo et al. (2022); Dixon et al. (2023); Bun et al. (2020); Ghazi et al. (2021) studied various notions of replicability in the context of learning tasks. In Bun et al. (2020), a notion called stability is introduced. The work of Impagliazzo et al. (2022), defined the notion of $\rho$-replicability. In Dixon et al. (2023), the authors introduce list replicability and certificate replicability. The work of Ghazi et al. (2021) studied list-global stability and pseudo-global stability. The works of Esfandiari et al. (2023); Karbasi et al. (2024); Eaton et al. (2023) studied replicability in the context of reinforcement learning and Bandits.

The work that is closely related to our work is the work of Esfandiari et al. (2023) in the context of MAB. Their definition of replicability for MAB is inspired by the notion of $\rho$-replicability of Impagliazzo et al. (2022). Informally, their definition requires that an algorithm for MAB is $\rho$-replicable if multiple runs of an algorithm, where all the runs share the internal random seed, will

result in the same trace, with probability greater than $1 - \rho$. Their goal was to design $\rho$-replicable algorithms for MAB with (near) optimal regret. Our definition of replicability is inspired by the list replicability notion of Dixon et al. (2023). In our work, the goal is to design list-replicable algorithms with *small* list size with (near) optimal regret. In this context, the notions of $\rho$ and list replicability are different as they do not imply each other. Thus our results are significantly different from that of Esfandiari et al. (2023). Since their algorithm's replicability depends on the random seeds/certificates, which could be exponentially many, their algorithm can have bad (exponential) list replicability when applied in our setting. Hence, different techniques are required to bound the list size.

## 5 MULTI-ARM BANDITS: UPPER BOUND

In this section, we introduce several list-replicable algorithms for the finite-arm setting. We first present an algorithm with no $T$ dependence yet with less ideal dependence on $k$. We will then attempt to improve the dependence on $k$ by sacrificing the dependence on other parameters.

### 5.1 A $(2^k, \delta)$-LIST REPLICABLE ALGORITHM

This algorithm is a standard phase-elimination type algorithm (yet our analysis for list-replicability is novel). For completeness, we present the full algorithm in Algorithm 1. The algorithm proceeds in batches (or rounds) and maintains a set of "good" arms. In each batch, the good arms are played uniformly in a lexicographic order to refine the estimation of their estimated means. At the completion of the batch, the arms whose estimated means are deemed non-optimal (even considering the estimation uncertainty) are removed. The batch length grows geometrically to provide more accurate estimates of the arms compared to the previous batch. The total number of batches is bounded by $O(\log T)$. Furthermore, the arms being played at every batch are guaranteed to satisfy certain optimality guarantees, and therefore, the overall regret per batch can be bounded with high probability.

---

**Algorithm 1:** $(2^k, \delta)$-List Replicable for MAB

---

**Input:** Horizon $T$, Actions $[k]$, Failure probability $\delta$
Number of batches: $B \leftarrow \lfloor \log_{25} T \rfloor$, Initial good arm set: $\mathcal{A}_1 \leftarrow [k]$
**for** *b=1,...,B* **do**

    Denote error parameter: $\epsilon_b \leftarrow \sqrt{\frac{ck}{5^{2b}} \log \frac{kB}{\delta}}$ for some constant $c > 0$,

    Set deletion criteria: $\mathfrak{D}_b \leftarrow 3\epsilon_b$

    For each arm $i \in \mathcal{A}_b$, play $i$ for $\left\lceil \frac{5^{2b}}{|\mathcal{A}_b|} \right\rceil$ times, and obtain an empirical mean $\hat{\mu}_{i,(b)}$ (halt the
    algorithm if $T$ arms have been played)

    For all $i \in \mathcal{A}_b$: let $\hat{\Delta}_{i,(b)} \leftarrow \max_{a \in \mathcal{A}_b} \hat{\mu}_{a,(b)} - \hat{\mu}_{i,(b)}$

    Eliminate bad arms: $\mathcal{A}_{b+1} \leftarrow \{i \in \mathcal{A}_b : \hat{\Delta}_{i,(b)} \leq \mathfrak{D}_b\}$.

**end**

---

#### 5.1.1 ANALYSIS

Our analysis for replicability relies on the novel observation that the total number of traces is determined by the number of possible batches for a sub-optimal arm to be eliminated. If each sub-optimal is eliminated in at most 2 different batches with high probability, then the overall list complexity is at most $2^k$. In particular, a sub-optimal arm is always eliminated in one of two consecutive batches, with high probability. More formally, the guarantee of the algorithm is presented as follows.

**Theorem 5.1.** *For every $\delta \in (0, 1]$, Algorithm 1 is $(2^k, \delta)$-list replicable, and for every input bandit instance satisfying Assumption 2.1, the algorithm enjoys a regret bound of $O\left( \sqrt{kT \log \frac{k \log T}{\delta}} \right)$ with probability at least $1 - \delta$.*

We provide the proof of the theorem, which is built on a sequence of lemmas and observations. Before we make our analysis formal, we recall our definition of the parameters. Note that $B =$

$\lfloor \log_{25} T \rfloor$ is the number of batches. For $b \in [B]$, note that $N_b = \left\lceil \frac{5^{2b}}{|\mathcal{A}|} \right\rceil$ is the number of pulls of each arm in the batch. Then each batch has a length of $T_b = |\mathcal{A}_b| \cdot N_b \geq 5^{2b}$. For $\epsilon_b = \sqrt{\frac{ck}{5^{2b}} \log \frac{kB}{\delta}}$, and $\hat{\mu}_{i,(b)}$ be the empirical mean of the arm $i$, we denote the following event $E_b$ : $\{(\forall i \in \mathcal{A}_b). \mu_i - \epsilon_b \leq \hat{\mu}_{i,(b)} \leq \mu_i + \epsilon_b\}$. Define the "good" event $E = \cap_{b \in [B]} E_b$. Then, by Hoeffding bound and a union bound, we have that $\Pr[E] \geq 1 - \delta$. Conditioned on $E$, we immediately have, $\forall b \in [B]$, $\mu^* - \epsilon_b \leq \max_{a \in \mathcal{A}_b} \hat{\mu}_{a,(b)} \leq \mu^* + \epsilon_b$ and $\Delta_i - 2\epsilon_b \leq \hat{\Delta}_{i,(b)} \leq \Delta_i + 2\epsilon_b$.

We now define a partition $\{I_1, \ldots, I_{B+1}\}$ on $[0,1]$. Let $\epsilon_0 = 1$ and $\epsilon_{B+1} = 0$. For each $m \in [B]$ define $I_m = (\epsilon_m, \epsilon_{m-1}]$ and $I_{B+1} = [\epsilon_{B+1}, \epsilon_B]$. Note for each arm $i \in [k]$, there exists a unique $m \in [B+1]$ such that $\Delta_i \in I_m$. We use $m_i$ to denote this unique partition number.

For $i \in [k]$, let $X_i$ be the random variable that denotes which batch arm $i$ is eliminated; we claim that conditioned on $E$, $X_i$ can take at most one of two values, namely $m_i$ or $m_i + 1$. This follows because of the geometric nature of $\epsilon_b$. The proof of the following can be found at the Appendix A.1.

**Lemma 5.2.** *For all $i \in [k]$, $\Pr(m_i \leq X_i \leq m_i + 1 | E) = 1$.*

To conclude that the algorithm is $(2^k, \delta)$-list replicable, consider the vectors of deletion sequences $(X_1, \ldots, X_k)$, with high probability see exactly these $2^k$ such vectors, each vector corresponds to a single unique trace. The proof of the following can be found in Appendix A.1. Thus concludes the proof of Theorem 5.1.

**Lemma 5.3.** *On $E$, the regret of Algorithm 1 satisfies $R_T \leq O\left(\sqrt{kT \log \frac{k \log T}{\delta}}\right)$.*

## 5.2 A $(O(k/\delta), \delta)$-List Replicable Algorithm

In this section, we present a novel modification to Algorithm 1 to further boost the list replicablility. For the above $(2^k, \delta)$-list replicablility, we obtained $2\epsilon_b$-estimates of $\Delta_i$ and eliminated those that were $3\epsilon_b$ away. The algorithm guarantees w.h.p. that each arm will be deleted randomly in one of two rounds. We shall now show that if we perturb the deletion criteria by a random shift picked at the start of the algorithm (while simultaneously getting better approximations for $\Delta_i$ by drawing more samples in each batch), then the arm will be deleted in exactly 1 round. Crucially, we show that if we pick at most $C = \left\lceil \frac{12k}{\delta} \right\rceil$ such perturbations, it results in at most $C$ many traces. The algorithm is formally presented in Algorithm 2.

---

**Algorithm 2:** $(12k/\delta, \delta)$-List Replicable for MAB

---

**Input:** Horizon $T$, Actions $[k]$, Failure probability $\delta$

Number of shifts: $C \leftarrow \left\lceil \frac{12k}{\delta} \right\rceil$, Number of batches: $B \leftarrow \left\lfloor \log_{25} \frac{T}{C^2} \right\rfloor$, Initial good arm set:

$\mathcal{A}_1 \leftarrow [k]$, Random shift: $r \sim \text{Unif}([C])$

**for** *b=1,…,B* **do**

    Denote error parameters: $\epsilon_b \leftarrow \sqrt{\frac{ck}{5^{2b}C^2} \log \frac{2kB}{\delta}}$, $\tau_b \leftarrow \frac{4C\epsilon_b}{3(\sqrt{2}-1)}$

    Set deletion criteria: $\mathfrak{D}_b \leftarrow \mathfrak{D}_{b,r} = 3\tau_b + (r-1) \cdot \frac{3(\tau_{b-1} - \tau_b)}{C}$

    For each arm $i \in \mathcal{A}_b$, play $i$ for $\left\lceil \frac{5^{2b}C^2}{|\mathcal{A}_b|} \right\rceil$ times, and obtain an empirical mean $\hat{\mu}_{i,(b)}$ (halt
    the algorithm if $T$ arms have been played)

    For all $i \in \mathcal{A}_b$: let $\hat{\Delta}_{i,(b)} \leftarrow \max_{a \in \mathcal{A}_b} \hat{\mu}_{a,(b)} - \hat{\mu}_{i,(b)}$

    Eliminate bad arms: $\mathcal{A}_{b+1} \leftarrow \{i \in \mathcal{A}_b : \hat{\Delta}_{i,(b)} \leq \mathfrak{D}_b\}$.

**end**

---

### 5.2.1 Analysis

Similar to the analysis of Algorithm 1, our new algorithm shares a similar structure of bounding the regret: the estimates of arms get increasingly improved, and only arms with better error guarantees will be played per batch. The challenging part is to show that the algorithm can only produce a small number of traces. The following theorem presents the formal guarantee.

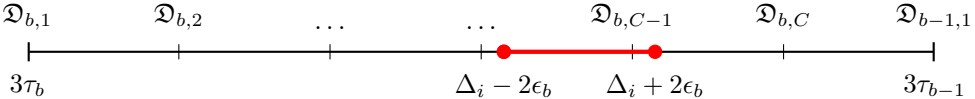

Figure 1: The deletion criteria $\mathfrak{D}_b$ is randomly chosen from $\mathfrak{D}_{b,1}, \ldots, \mathfrak{D}_{b,C}$

**Theorem 5.4.** *For every $\delta \in (0,1]$, Algorithm 2 is $\left(\frac{12k}{\delta}, \delta\right)$-list replicable, and for every input bandit instance satisfying Assumption 2.1, the algorithm enjoys a regret bound of $O\left(\frac{k}{\delta} \cdot \sqrt{kT \log \frac{k \log T}{\delta}}\right)$ with probability at least $1 - \delta$.*

We now present a proof of the theorem. Recall that $C = \lceil 12k/\delta \rceil$ the number of shifts, $B = \lfloor \log_{25} \frac{T}{C^2} \rfloor$. Let $T_b = 5^{2b}C^2$ and $N_b = \lceil \frac{T_b}{k} \rceil$, we draw $T_b$ samples each round, each arm gets $\geq N_b$ samples. Let $\epsilon_b = \sqrt{\frac{ck}{5^{2b}C^2} \log \frac{2kB}{\delta}}$ (for some $c > 0$) and $\tau_b = \frac{4C\epsilon_b}{3(\sqrt{2}-1)}$ and thus the estimate at each round is $\epsilon_b$-accurate with probability at least $1 - \delta$. For $b \in [B]$, and for $r \in [C]$, consider the deletion criterias in the algorithm: $\mathfrak{D}_{b,r} = 3\tau_b + (r-1) \cdot \frac{3(\tau_{b-1} - \tau_b)}{C}$. For a randomly picked $r \in [C]$, the deletion criteria at round $b$, denoted by the random variable $\mathfrak{D}_b$ is uniformly picked from $\mathfrak{D}_{b,1}, \ldots, \mathfrak{D}_{b,C}$. Note that the same $r$ is shared across all rounds.

Unlike in Algorithm 1, the decision boundary, $\mathfrak{D}_b \propto (C\epsilon_b)$ is much larger than the estimation error $\epsilon_b$. Hence, suppose an arm $i$ has a gap $\Delta_i \in [\mathfrak{D}_b, \mathfrak{D}_{b-1}]$ and is more than $O(\epsilon_b)$ from the boundary of the interval, the algorithm will eliminate it with high probability in batch $b$. Only the "bad" arms whose gaps are within $O(\epsilon_b)$ to the boundary has the possibilities to be not eliminated in batch $b$. We will show that for the $C$ we choose, with certain probability no bad arms exists and thus there is unique trace from the algorithm. Below, we make this intuition more formal.

**Observation 5.5.** *For all $b \in [B]$ and $r \in [C]$, $3\tau_b \leq \mathfrak{D}_{b,r} < 3\tau_{b-1}$. In particular, this always holds for the random variable $\mathfrak{D}_b$, $3\tau_b \leq \mathfrak{D}_b < 3\tau_{b-1}$.*

Analogous to the analysis of Algorithm 1, for $b \in [B]$, we define the event as $E_b : (\forall i \in \mathcal{A}_b). \mu_i - \epsilon_b \leq \hat{\mu}_{i,(b)} \leq \mu_i + \epsilon_b$, and the "good" event $E = \cap_{b \in [B]} E_b$. Via Hoeffding and a union bound, we have for all $b \in [B]$. $\Pr(E_b) \geq 1 - \frac{\delta}{2B}$ and $\Pr(E) \geq 1 - \frac{\delta}{2}$. Define the partition $\{I_1, \ldots, I_{B+1}\}$ on $[0,1]$. Let $\tau_0 = 1$ and $\tau_{B+1} = 0$. For each $m \in [B]$ define $I_m = (\tau_m, \tau_{m-1}]$ and $I_{B+1} = [\tau_{B+1}, \tau_B]$. For each arm $i \in [k]$, there exists a unique $m \in [B+1]$ such that $\Delta_i \in I_m$. We use $m_i$ to denote this unique partition number.

We now claim that each arm $i$ will be eliminated in one of three batches, namely $m_i, m_i+1, m_i+2$, instead of the two in the case of the previous algorithm. Intuitively, an arm that was supposed to be deleted in the second round might be delayed to the third round because of the perturbation in the deletion criteria. The full proof can be found in Appendix A.2.

**Lemma 5.6.** *For all $i \in [k]$, $\Pr(m_i \leq X_i \leq m_i + 2 | E) = 1$.*

Referring to Figure 1, observe that if $r = C - 1$ were picked, it would be a "bad" because even conditioned on $E$, sometimes the estimate of $\Delta_i$ lies on the left of the deletion criteria, $\mathfrak{D}_b$ and other times $\Delta_i$ lies on the right of the $\mathfrak{D}_b$, i.e., the round the arm is deleted is inconsistent even when estimates are good. The other $r$s are "good" because the arm consistently either gets eliminated this round $b$ or its deletion gets postponed to a future round.

**Definition 5.7.** *Let $i \in [k]$, $r \in [C]$, and $b \in [B]$. We say,*

1. *$r$ is b-bad for $i$ if $\mathfrak{D}_{b,r} \in [\Delta_i - 2\epsilon_b, \Delta_i + 2\epsilon_b]$.*

2. *$r$ is bad for $i$ if it is b-bad for $i$ for some $b = m_i, m_i + 1, m_i + 2$.*

3. *$r$ is bad if it is bad for $i$ for some $i \in [k]$. Otherwise, $r$ is said to be good.*

$\mathfrak{D}_{b,r}$s were carefully chosen so that every batch $b \in [B]$ the length of the approximation interval of $\Delta_i$ at round $b$ is exactly equal to the distance between consecutive shifts of the same round $\mathfrak{D}_{b,r}$ and $\mathfrak{D}_{b,r+1}$. As a consequence, there can only be at most two $\mathfrak{D}_{b,r}$s contained in the interval

$[\Delta_i - 2\epsilon_b, \Delta_i + 2\epsilon_b]$. With some rudimentary counting, the following lemma holds whose full proof can be found in Appendix A.2.

**Lemma 5.8.** *There are at most $6k$ bad $r$s.*

When a good $r$ is chosen, we see exactly one trace (conditioned on $E$), because the deletion criteria is always outside the estimation interval for all arms. Thus if we uniformly at random pick an $r \sim \text{Unif}[C]$, then with probability at most $\frac{6k}{C} = \frac{\delta}{2}$, $r$ is bad. Therefore, the number of traces is at most $C = \frac{12k}{\delta}$ when both these events occur. The probability that both estimates are good and $r$ is good are independent and is bounded by: $\Pr(E \cap r \text{ is good}) \geq \left(1 - \frac{\delta}{2}\right)^2 \geq 1 - \delta$. The proof of the following lemma is similar to the regret analysis of the previous section and can be found in Appendix A.2. Thus concludes the proof of Theorem 5.4.

**Lemma 5.9.** *On $E$ and the event that $r$ is good, the regret of Algorithm 2 satisfies, $R_T \leq O\left(\frac{k}{\delta}\sqrt{kT \log \frac{k \log T}{\delta}}\right)$.*

## 5.3 A $(k^{O(1)}, \delta)$-LIST REPLICABLE ALGORITHM WITH $k^{3/2}T^{1/2+O(1)}$-REGRET

In this section, we refine the tradeoff between list-replicability and the regret bound. We rely on the following critical routine proposed in Dixon et al. (2023).

**Theorem 5.10** (Dixon et al. (2023), Replicable Mean Estimation). *Let $\epsilon, \delta \in (0, 1]$ and $k \in \mathbb{N}$. For any random vector $V \in \mathbb{R}^k$, suppose there exists an algorithm takes samples from $V$ and produces an estimator $\widehat{V}$ such, that $\|\mathbb{E}[V] - \widehat{V}\|_\infty \leq \frac{\epsilon}{2k}$ with probability at least $1 - \delta$. Then there exists an algorithm, `ReplicableMeans`, which takes $\widehat{V}$ as an input and output a $((k+1), \delta)$-list replicable estimator $\overline{V}$, which satisfies $\|\mathbb{E}[V] - \overline{V}\|_\infty \leq \epsilon$, with probability at least $1 - \delta$.*

We shall argue about replicability based on the subset of arms that are eliminated. Let $B > 1$ be an integer, denoting the number of batches, and $\delta \in (0, 1]$. Let $\nu \approx \frac{1}{2^{B+1}-2}$ with $\alpha = \frac{1}{2} + \nu$. We shall demonstrate how to create a $((k+1)^{B-1}, \delta)$-list replicable algorithm with $\tilde{O}(kT^\nu\sqrt{kT})$-regret. The algorithm is a constant batched algorithm where $B$ is the number of batches. In each batch, bad arms are deleted in a replicable manner with the help of Theorem 5.10; thus, in each batch, only $(k+1)$ possible subset of arms are deleted (instead of the naïve $2^k$). Via rule of product, over $B - 1$ batches $(k+1)^{B-1}$ traces are observed. Note that the eliminations in batch $B$ does not affect the replicability. Therefore the replicability is $(k+1)^{B-1}$. The algorithm is formally presented in Algorithm 3.

---

**Algorithm 3:** $((k+1)^{B-1}, \delta)$-List Replicable for MAB

**Input:** Horizon $T$, Actions $[k]$, Number of Batches $B > 1$, Failure probability $\delta$

Regret paramter: $\nu \leftarrow \frac{1}{2^{B+1}-2}$, $\alpha \leftarrow \frac{1}{2} + \nu$, Initial good arm set: $\mathcal{A}_1 \leftarrow [k]$

**for** *b=1,...,B* **do**

> For each arm $i \in \mathcal{A}_b$, play $i$ for $N_b = \left\lceil T^{\alpha\left(2 - \frac{1}{2^{b-1}}\right)} / |\mathcal{A}_b| \right\rceil$ times and obtain an empirical mean $\hat{\mu}_{i,(b)}$ (the algorithm halts if it has played $T$ arms)
>
> Use Theorem 5.10: $\langle \hat{\mu}_{i,(b)} \rangle_{i \in \mathcal{A}_b} \leftarrow \text{ReplicableMeans}\left(\langle \hat{\mu}_{i,(b)} \rangle_{i \in \mathcal{A}_b}\right)$
>
> For all $i \in \mathcal{A}_b$: let $\hat{\Delta}_{i,(b)} \leftarrow \max_{a \in \mathcal{A}_b} \hat{\mu}_{a,(b)} - \hat{\mu}_{i,(b)}$
>
> Let $T_b = N_b|\mathcal{A}_b|$ be the total steps of batch $b$
>
> Denote error parameters: $\epsilon_b \leftarrow \sqrt{\frac{ck^3}{T_b} \log \frac{kB}{\delta}}$
>
> $\mathcal{A}_{b+1} \leftarrow \{i \in \mathcal{A}_b : \hat{\Delta}_{i,(b)} \leq 2\epsilon_b\}$.

**end**

---

### 5.3.1 ANALYSIS

**Theorem 5.11.** *For every $\delta \in (0,1]$, and integer $B > 1$, Algorithm 3 is $\left((k+1)^{B-1}, \delta\right)$-list replicable, and for every input bandit instance satisfying Assumption 2.1, the algorithm enjoys a regret bound of $O\left(kT^{\nu}\sqrt{kT \log \frac{kB}{\delta}}\right)$ with probability at least $1 - \delta$, where $\nu = \frac{1}{2^{B+1}-2}$.*

**Remark 5.12.** *We note that when $B = 2$, the algorithm is $(k+1, \delta)$-list replicable while enjoys a regret bound of $\widetilde{O}(k^{3/2}T^{2/3})$. For $B = \log \log T$, the algorithms enjoys a regret bound of $\widetilde{O}(k^{3/2}\sqrt{T})$, while with $k^{\log \log T}$ list-complexity Cesa-Bianchi et al. (2013).*

We present the proof of Theorem 5.11. For $b \in [B]$, let $T_b = \Theta(T^{\alpha(2 - \frac{1}{2^{b-1}})})$ be the batch length. Note that if $\nu = \frac{1}{2^{B+1}-2}$ then $T_B = T$. Therefore $\nu \approx \frac{1}{2^{B+1}-2}$ is sufficient to guarantee $\sum_{b \in [B]} T_b \approx T$. Let $\epsilon_b = \sqrt{\frac{ck^3}{T_b} \log \frac{kB}{\delta}}$ for some $c$. Therefore, for each $b \in [B]$, with probability at least $1 - \delta/B$, for each $i \in \mathcal{A}_b$, $|\hat{\mu}_{i,(b)} - \mu_i| \leq \frac{\epsilon_b}{2k}$. Therefore, by Theorem 5.10, `RelicableMeans` outputs a $(k+1, \delta/B)$-list means with error at most $\epsilon_b$ to each arm.

We now argue Algorithm 3 is $((k+1)^{B-1}, \delta)$-list replicable. Due to the list replicability of the mean estimators, we see at most $(k+1)$ possible subset deletions at each $b = 1, \ldots, B - 1$ with high probability. Therefore, we see at most $(k+1)^{B-1}$ traces with probability $\geq 1 - \delta$. The proof of the following lemma similar to the previous regret analysis, and the proof can be found in Appendix A.3. Thus concludes the proof of Theorem 5.11.

**Lemma 5.13.** *The regret of Algorithm 3 with probability at least $1 - \delta$ is bounded by $R_T \leq O\left(kT^{\nu}\sqrt{kT \log \frac{kB}{\delta}}\right)$.*

## 6 MULTI-ARM BANDITS: LOWER BOUND

In this section, we prove that any algorithm for $k$-MAB that has a regret $o(T)$ with high probability can not be $(k-1)$ list replicable.

**Theorem 6.1.** *There is no $(k-1, \delta)$-list replicable algorithm for $k$-MAB that has $o(T)$ regret with probability $1 - \delta$, for $\delta \leq \frac{1}{k+1}$.*

The proof of this lower bound theorem makes use of a cubical version of Sperner/KKM lemma De Loera et al. (2002). Informally, Sperner/KKM lemma says the following: if we "properly color" a $d$ dimensional cube with $d + 1$ colors, then there is a point $p$ in the cube so that any arbitrary small neighborhood of $p$ contains $d + 1$ points with different colors.

We prove Theorem 6.1 in two steps. Step (1): we establish that any $(k-1)$-list replicable algorithm with $o(T)$ regret yields a $(k-1)$ list replicable algorithm for a problem known as $(k, \epsilon, \delta)$-BESTARM. Step (2): we establish that $(k, \epsilon, \delta)$-BESTARM does not admit $(k-1)$-list replicable algorithm.

We provide a proof sketch after stating relevant definitions and necessary results. The detailed proofs are given in Appendix A.4.1. We say an arm $a$ is an $\epsilon$-*best arm* if $\mu^* - \mu_a < \epsilon$.

**Definition 6.2** ($(k, \epsilon, \delta)$-BESTARM). *Let $k \in \mathbb{N}$ and $\epsilon, \delta \in (0, 1)$. The $(k, \epsilon, \delta)$-BESTARM is following problem: given $k$ arms with unknown means $\mu_1, \ldots, \mu_k$. Output an $\epsilon$-best arm. with probability $\geq 1 - \delta$. The algorithm can sample from the distribution of the arms.*

**Definition 6.3** (List replicable algorithms). *An algorithm $\mathcal{A}$ for $(k, \epsilon, \delta)$-BESTARM is $\ell$-list replicable if there is a list $L \subseteq [k]$ of size $\ell$ so that (1) for every $i \in L$, $\mu^* - \mu_i < \epsilon$ (2) $\Pr[\mathcal{A} \text{ outputs a arm from } L] \geq 1 - \delta$.*

**Lemma 6.4.** *If there is an $(\ell, \delta)$-list replicable algorithm for $k$-MAB that has $o(T)$ regret with probability $\geq 1 - \delta$ then there is a $\ell$-list replicable algorithm for $(k, \epsilon, 2\delta)$-BESTARM for any constant $\epsilon$ and $\delta$.*

The above Lemma formalizes Step (1). The proof of the lemma is based on the observation that any $o(T)$-regret algorithm for $k$-MAB should not play a bad arm too frequently. Thus, we could use $k$-MAB algorithm with $o(T)$ regret to tell us an approximate best arm by simply looking at the

most frequently played arm. Therefore, to prove Theorem 6.1 it suffices to show that there is no $(k-1, \frac{1}{k+1})$-list replicable algorithm for $(k, \epsilon, \delta)$-BESTARM (Step (2)).

Following is a sketch of the proof. Suppose there is a $(k-1, \delta)$-list replicable algorithm $\mathcal{A}$ for $(k, \epsilon, \delta)$-BESTARM. Let $\mathcal{C}$ be a $(k-1)$-dimensional cuboid. For each point $\vec{p} = (\alpha_1, \ldots, \alpha_{k-1}) \in \mathcal{C}$, we associate $k$ arms with means $(\alpha_1, \ldots, \alpha_{k-1}, \frac{1}{2})$. Each point $\vec{p} \in \mathcal{C}$ is assigned a color from $[k]$ as follows: Consider the behavior of the algorithm $\mathcal{A}$ on the $k$ arms associated with the point $\vec{p}$. If arm $i \in [k]$ is the most likely output of $\mathcal{A}$, then color the point $\vec{p}$ with color $i$.

It can be shown that the above coloring is a Sperner/KKM coloring of $\mathcal{C}$. Thus by Sperner/KKM lemma, there exists a point $\vec{p} \in \mathcal{C}$, such that for every $\nu > 0$, the $\nu$-neighborhood of $p$ (points that are $\nu$ close in $\ell_\infty$-norm sense) contains $k$ points $\vec{p}_1, \ldots, \vec{p}_k$ with distinct colors. Without loss of generality, let the most likely output of $\mathcal{A}$ on the arms associated with the point $\vec{p}_i$ be $i$. The crucial observation is that since $\mathcal{A}$ is $(k-1, \delta)$-list replicable, the probability of the most likely output is at least $\frac{1-\delta}{k-1}$. On the other hand, since the points $\vec{p}_1, \ldots, \vec{p}_k$ are arbitrarily close, by data processing inequality, it can be argued that the behavior of the algorithm $\mathcal{A}$ must be very similar on each of these inputs. This would imply that the algorithm $\mathcal{A}$ on instance associated with $\vec{p}_1$ must output each arm in $[k]$ with probability at least $\frac{1-\delta}{k-1}$. This leads to contradictions as the sum of these probabilities is more than 1, for the choice of $\delta$.

**Remark.** Consider Algorithm 3 with $B = 2$. With a slight modification (play the best estimate arm in the second batch), this is a $(k, 0)$-list replicable algorithm with regret $\tilde{O}(T^{2/3})$. Thus, by Theorem 6.1, in the sub-linear regret regime, the list size of $k$ is optimal. We also point out that $\delta \leq \frac{1}{k+1}$ is necessary due to the following: Consider the $(k, 0)$-list replicable algorithm as discussed. Since there are only $k$ traces, there is a trace with probability $\leq \frac{1}{k}$. The remaining $k-1$ traces have probability at least $(1 - \frac{1}{k})$. Thus this is a $(k-1, \frac{1}{k})$ list-replicable algorithm.

# 7 LINEAR BANDITS

In this section, we provide the algorithm for list-replicable linear bandit. In the linear bandit setting, we are given a set of actions $\mathcal{A} \subset \mathbb{R}^d$ for some dimension $d$. Similar to the MAB setting, an agent is allowed to take one action $a_t$ from $\mathcal{A}$ at each time $t$ and receive a reward $r_t$. The assumption is that $r_t$ satisfies a linear structure.

## 7.1 ALGORITHM

The algorithm takes a phase-elimination batched structure, in each batch $b$, the algorithm maintains a set of surviving actions $\mathcal{A}_b$ such that for all $a \in \mathcal{A}_b$, $\mu_{a^*} \leq \mu_a + \epsilon_{b-1}$ for some $\epsilon_{b-1}$ depending on the previous batch. Given a target regret bound $R_T$, we choose the current batch size, $T_b$, so that $\epsilon_{b-1} T_b \lesssim R_T$ and $\sum_{b=1}^{B} T_b \lesssim T$.

Similar to the MAB analysis, $B$ is either constant or log-logarithmic in $T$. The critical component to maintain the regret bound is that we need to eliminate the arms with errors smaller than previous batch. We first choose a small set of arms from the survival set $\mathcal{A}_b$. These arms are chosen such that they can be "good" representatives (in terms of linear regression) of the survival set. More specifically, we choose the $G$-optimal design (specified shortly), which contains $2d$ arms. Our algorithm plays each arm a specific number of times to obtain a sufficient number of samples about its mean. With these samples, we call the `ReplicableMeans` (Theorem 5.10) estimator to produce a $(2d + 1)$-list-replicable for their means. We then solve linear regression for these estimated means to obtain $\theta^*$ for this batch and eliminate the arms with sub-optimality below the accuracy level. We will show that the algorithm halts in at most $B$ batches and achieves the desired regret bound. The list-complexity is therefore $(2d + 1)^{B-1}$.

## 7.2 ANALYSIS

**Definition 7.1.** *For any compact set $\mathcal{A} \subset \mathbb{R}^d$, let $\pi : \mathcal{A} \to [0, 1]$ be a distribution on $\mathcal{A}$. Denote $V(\pi) = \sum_a \pi(a) a a^\top$ and $g(\pi) = \sup_{a \in \mathcal{A}} \lim_{\lambda \to 0} \|a\|_{(\lambda I + V(\pi))^{-1}}$. $\pi$ is a c-G-optimal design for $\mathcal{A}$, if $g(\pi) \leq cd$.*

---

**Algorithm 4:** $((2d+1)^{B-1}, \delta)$-List Replicable Linear Bandit Algorithm

---

**Input:** Horizon $T$, Number of batches $B > 1$, Failure probability $\delta > 0$

Initialize: $\mathcal{A}_1 = \mathcal{A}$, $\epsilon_0 = 1$, $\nu \leftarrow \frac{1}{2^{B+1}-2}$, $\alpha \leftarrow \frac{1}{2} + \nu$

**for** *b=1,...,B* **do**

> Find a $\sqrt{2}$-G-optimal design $\pi_b$ for $\mathcal{A}_b$ using Corollary 7.3 with $d' = |\mathrm{supp}(\pi_b)| \le 2d$.
>
> For $a \in \mathrm{supp}(\pi_b)$, play $a$ for $N_b \leftarrow \lceil T^{\alpha(2-\frac{1}{2^{b-1}})}/d' \rceil$ times and obtain empirical mean
> $\mu'_{(b)} = (\mu'_{a,(b)})_{a \in \mathrm{supp}(\pi_b)}$ (halt the algorithm if $T$ arms are played)
>
> Let $\bar{\mu}_{(b)} \leftarrow \texttt{ReplicableMeans}(\mu'_{(b)})$
>
> Let $\Phi_b = [a]^\top_{a \in \mathrm{supp}(\pi_b)}$ be the matrix of stacking all actions in $\mathrm{supp}(\pi_b)$
>
> Solve for estimated mean by: $\hat{\theta} \leftarrow \left(\Phi_b^\top \Phi_b\right)^\dagger \Phi_b^\top \bar{\mu}_{(b)}$
>
> Denote error parameter: $\epsilon_b \leftarrow \sqrt{\frac{cd^4 \log \frac{d'B}{\delta}}{T_b}}$, where $T_b = |\mathrm{supp}(\pi_b)|N_b$
>
> Eliminate actions $a \in \mathcal{A}_b$ with sub-optimality below the accuracy level:
> $\mathcal{A}_{b+1} \leftarrow \{a \in \mathcal{A}_b : \max_{a' \in \mathcal{A}_b}\langle a', \hat{\theta}\rangle - \langle a, \hat{\theta}\rangle \le \epsilon_b/2\}$

**end**

---

**Theorem 7.2** (G-optimal design Kiefer & Wolfowitz (1960)). *For any compact subset $\mathcal{A} \subset \mathbb{R}^d$, there exists a 1-G-optimal design $\pi$, $G \subset \mathcal{A}$ such that $|\mathrm{supp}(\pi)| = d$.*

**Corollary 7.3.** *For any compact subset $\mathcal{A} \subset \mathbb{R}^d$, there exists a $\sqrt{2}$-G-optimal design $\pi$, $G \subset \mathcal{A}$, such that $|\mathrm{supp}(\pi)| \le 2d$ and $\pi$ is a uniform distribution over $\mathrm{supp}(\pi)$.*

**Theorem 7.4** (Guarantee of Algorithm 4). *For every $\delta \in (0,1]$, and integer $B > 1$, Algorithm 4 is $\left((2d+1)^{B-1}, \delta\right)$-list replicable, and for every input bandit instance satisfying Assumption 2.4, enjoys a regret bound of $O\left(d^2 T^\nu \sqrt{T \log \frac{dB}{\delta}}\right)$ with probability at least $1 - \delta$, where $\nu = \frac{1}{2^{B+1}-2}$.*

## 8 CONCLUSION

This work introduces and initiates the study of list replicability for the Multi-Armed Bandits problem. We designed various list-replicable algorithms for $k$-MAB with near-optimal regret in $T$ and list-complexity linear/polynomial in $k$. To complement these results, we show that we can not hope to design $(k-1)$-list replicable algorithms with sublinear regret. An open question is to close the gap between upper and lower bounds on the list complexity. For example, in Theorem 5.4, if we chose $\delta = \frac{1}{k}$, then the list size is $O(k^2)$. Can we prove that list complexity is $\Omega(k^2)$? A general question is to investigate trade-offs between list complexity and regret.

**Remark about UCB.** The list complexity of UCB-based algorithms is exponentially dependent on $T$. Consider the case when $k = 2$ and both arms have identical distributions. Let $\mathrm{UCB}_1$ and $\mathrm{UCB}_2$ represent the UCB estimates of arm 1 and arm 2, respectively. Let us assume that at some time $t$, $\mathrm{UCB}_1 < \mathrm{UCB}_2$. UCB algorithm dictates that we shall play arm 2 till the estimate of arm 1 is larger. However, because the samples are random, the time at which we start playing arm 1 is probabilistic. For each time step where a switch is possible results in a new trace. This trait makes the list complexity of UCB-based algorithms exponentially dependent on $T$. Note that the list complexities of our algorithms are independent of $T$. We leave it as an open question about what changes need to be made to UCB to guarantee list replicability.

**Acknowledgements.** Pavan and Michael's work is partly supported by NSF grants 2130536 and 234225. Vinod's work is partly supported by NSF grants 2130608 and 2342244.

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

## A  APPENDIX / SUPPLEMENTAL MATERIAL

### A.1  PROOFS OF SECTION 5.1

We first note the following observation about Algorithm 1

**Observation A.1.** $\epsilon_{b-1} = 5\epsilon_b$ and $\epsilon_{B+1} < \epsilon_B < \ldots < \epsilon_{b+1} < \epsilon_b < \epsilon_{b-1} \ldots < \epsilon_1 < \epsilon_0$

*Proof of Lemma 5.2.* We first prove for all $i \in [k]$, $\Pr(X_i \geq m_i | E) = 1$

Let $i \in [k]$, for each $b \leq m_i - 1$, $\hat{\Delta}_{i,(b)} \leq \Delta_i + 2\epsilon_b \leq \epsilon_{m_i-1} + 2\epsilon_b \leq \epsilon_b + 2\epsilon_b = \mathfrak{D}_b$. This is to say that for rounds $b$ such that $b \leq m_i - 1$, the highest estimate of $\Delta_i$ is no more than the deletion criteria $\mathfrak{D}_b$. Thus, $i$ will not be deleted.

We now prove for all $i \in [k]$, $\Pr(X_i \leq m_i + 1 | E) = 1$

Let $i \in [k]$ and $b = m_i + 1$, we have $\hat{\Delta}_{i,(b)} \geq \Delta_i - 2\epsilon_b > \epsilon_{m_i} - 2\epsilon_b = \epsilon_{b-1} - 2\epsilon_b = 3\epsilon_b = \mathfrak{D}_b$. This is to say that the lowest estimate of $\Delta_i$ at round $m_i + 1$, is strictly more than the deletion criteria $\mathfrak{D}_b$, therefore $i$ will be eliminated at the latest by round $m_i + 1$. $\square$

*Proof of Lemma 5.3.* Observe that the regret in each round $b \in \{2, \ldots, B\}$ is at most $T_b \mathfrak{D}_{b-1}$ and for round 1 the expected regret is at most $T_1 \cdot 1$

$$R_T \leq T_1 \cdot 1 + \sum_{b=2}^{B} T_b \mathfrak{D}_{b-1} = 25 + 3 \sum_{b=2}^{B} 5^{2b} \cdot \sqrt{\frac{ck}{5^{2(b-1)}} \log \frac{1}{\theta}}$$

$$= 25 + 3\sqrt{25ck \log \frac{1}{\theta}} \sum_{b=2}^{B} 5^b \leq 25 + 3\sqrt{25ck \log \frac{1}{\theta}} \cdot 25^{\frac{B}{2}}$$

$$= O\left(\sqrt{kT \log\left(\frac{k \log T}{\delta}\right)}\right)$$

$\square$

### A.2  PROOFS OF SECTION 5.2

*Proof of Lemma 5.6.* Following a similar analysis for the proof of Lemma 5.2

We first prove, for all $i \in [k]$, $\Pr(X_i \geq m_i | E) = 1$.

Let $i \in [k]$, for each $b \leq m_i - 1$

$$\hat{\Delta}_{i,(b)} \leq \Delta_i + 2\epsilon_b \leq \tau_{m_i-1} + 2\epsilon_b \leq \tau_b + \frac{3(\sqrt{2}-1)}{2C}\tau_b \leq 3\tau_b \leq \mathfrak{D}_b$$

We now prove, For all $i \in [k]$, $\Pr(X_i \leq m_i + 2|E) = 1$.

Note that $\epsilon_b < \tau_b$. Let $i \in [k]$, for $b = m_i + 2$,

$$\hat{\Delta}_{i,(b)} \geq \Delta_i - 2\epsilon_b > \tau_{m_i} - 2\tau_b = 5\tau_{b-1} - 2 \cdot \frac{\tau_{b-1}}{5} > 3\tau_{b-1} > \mathfrak{D}_b$$

$\square$

*Proof of Lemma 5.9.* Ignoring constants.

$$R_T \leq T_1 \cdot 1 + \sum_{b=2}^{B} T_b \mathfrak{D}_{b-1} \leq \sum_{b=2}^{B} 5^{2b} C^2 \cdot \sqrt{\frac{k}{5^{2b}} \log \frac{1}{\theta}} = C^2 \sqrt{k \log \frac{1}{\theta}} \cdot 25^{\frac{B}{2}} = \frac{k}{\delta} \sqrt{kT \log \frac{k \log T}{\delta}}.$$

$\square$

## A.3 PROOFS OF SECTION 5.3

*Proof of Lemma 5.8.* We first prove

**Claim A.2.** *Let $i \in [k]$ and $b = m_i, m_i + 1, m_i + 2$, there are at most two b-bad rs for i.*

*Proof.* Let $b = m_i, m_i + 1, m_i + 2$. Firstly, observe that the decision boundaries $\mathfrak{D}_{b,1}, \mathfrak{D}_{b,2}, \ldots, \mathfrak{D}_{b,C}$ are spaced by $4\epsilon_b$. The estimate interval $[\Delta_i - 2\epsilon_b, \Delta_i + 2\epsilon_b]$ has length $4\epsilon_b$ and can contain at most two $\mathfrak{D}_{b,r}, \mathfrak{D}_{b,r+1}$. See Figure 1. $\square$

Let $i \in [k]$. Fix $b = m_i$ or $m_i + 1$ or $m_i + 2$, the number of $b$-bad $r$'s for $i$ at most 2 via Claim A.2. In the worst case, there are 2 bad $rs$ for each of $m_i, m_i + 1$, and $m_i + 2$, and all of them are different. Therefore, there can be at most $3 \times 2 = 6$ for $i$. Therefore, there can be at most $k \times 6$ bad $rs$. $\square$

*Proof of Lemma 5.13.* Let $\theta = \frac{\delta}{B}$, we have, ignoring constants

$$R_T \leq T_1 \cdot 1 + \sum_{b=2}^{B} T_b \cdot \epsilon_b = T^{\alpha} + \sum_{b=2}^{B} T_b \cdot \sqrt{\frac{k^3}{T_{b-1}} \log \frac{k}{\theta}} = T^{\alpha} + \sqrt{k^3 \log \frac{k}{\theta}} \sum_{b=2}^{B} \frac{T_b}{\sqrt{T_{b-1}}}$$

$$= T^{\alpha} + \sqrt{k^3 \log \frac{k}{\theta}} \sum_{b=2}^{B} T^{\alpha} = T^{\alpha} + BT^{\alpha} \sqrt{k^3 \log \frac{kB}{\delta}}$$

$$= O\left( kT^{\nu} \sqrt{kT \log \frac{kB}{\delta}} \right)$$

$\square$

## A.4 PROOFS OF SECTION 6

*Proof of Lemma 6.4.* Let $\mathcal{M}$ be a $(\ell, \delta)$-list replicable algorithm for $k$-MAB that has a regret $o(T)$ with probability $\geq 1 - \delta$. We design a $\ell$-list replicable algorithm $\mathcal{A}$ for $(k, \epsilon, 2\delta)$-BESTARM using $\mathcal{M}$. $\mathcal{A}$ works as follows: Simulate $\mathcal{M}$ for $T$ steps and let $N_i$ be the number of times arm $i$ was played. Output the arm that is most frequently played: That is, output $\arg\max_i N_i$, breaking ties by outputting the smallest numbered arm. The list replicability of $\mathcal{A}$ follows due to the $\ell$-list replicability $\mathcal{M}$. It suffices to show that $\mathcal{A}$ outputs an arm $\epsilon$-close to the best arm with probability $\geq 1 - \delta$.

Let $\Omega$ be the sample space of the algorithm $\mathcal{M}$. Each sample point $\omega \in \Omega$ is a sequence of alternating actions and rewards. We say that a sample point $\omega$ is *good* if the regret of $\mathcal{M}$ along $\omega$ is $o(T)$ and the trace associated with $\omega$ is in the list. By union, it follows that $\Pr[\omega \text{ is good}] \geq 1 - 2\delta$.

Consider a good $\omega \in \Omega$ Let $a = \mathcal{A}(\omega)$ be the output in this sample point. We now claim the $\mu_a \geq \mu^* - \epsilon$. Assume for contradiction's sake, that the output $\mu_a < \mu^* - \epsilon$. We know that $N_a(\omega) \geq \frac{T}{k}$ because $\mathcal{A}$ outputs most frequently played arm. Hence the regret along $\omega$ is $\geq N_a(\omega)\epsilon \geq \frac{T\epsilon}{k}$. Since $\epsilon$ and $k$ are constants, this is a contradiction. Thus $\Pr[\mathcal{A} \text{ outputs } \epsilon\text{-best arm}] \geq 1 - \delta$. $\square$

### A.4.1 LOWERBOUND ON THE LIST COMPLEXITY OF $(k, \epsilon, \delta)$-BESTARM

In this section, we will prove that there is no $(k - 1, \frac{1}{k+1})$-list replicable algorithm for $(k, \epsilon, \delta)$-BESTARM. We first introduce the necessary definitions and notation and then state Sperner/KKM Lemma.

**Definition A.3** (Sperner/KKM Coloring). *Let $d \in \mathbb{N}$ and let $\mathcal{C}$ be a $d$-dimensional cuboid and let $\mathcal{V}$ be the vertex set (corners) of $\mathcal{C}$. For a face $F$ of $\mathcal{C}$ (of any dimension) let $\mathcal{V}_F$ be the set of vertices of $F$. A coloring function color $: \mathcal{C} \to [d + 1]$ of $C$ is called a Sperner/KKM coloring if (1) $\{color(v) \mid v \in \mathcal{V}\} = [d + 1]$ (intuitively, all colors are used on the corners) (2) for any face $F$ of $\mathcal{C}$, for any point $p \in F$, it holds that $color(p) = color(v)$ for some $v \in \mathcal{V}_F$ (informally, the color of $p$ is color of one of the vertices in the face $F$).*

**Theorem A.4** (Cubical Sperner/KKM lemma De Loera et al. (2002)). *Let $d \in \mathbb{N}$ and let $\mathcal{C}$ be a $d$-dimensional cuboid. Let color $: \mathcal{C} \to [d + 1]$ be a Sperner/KKM coloring of $\mathcal{C}$. Then there is a point $p$ such that for every $\nu > 0$, $\nu$-neighbourhood of $p$, $\mathcal{B}_\nu^\infty(p)$ that contains $d + 1$ points with distinct colors.*

**Theorem A.5.** *Let $\epsilon \leq \frac{1}{2k}$ and $\delta \leq \frac{1}{k+1}$. There is no $(k - 1)$-list replicable algorithm for $(k, \epsilon, \delta)$-BESTARM.*

*Proof of Theorem A.5.* Let $\mathcal{A}$ be a $(k-1)$-list replicable algorithm for the $(k, \epsilon, \delta)$-BESTARM where the total number of times arms are played is $n$.

We will use Sperner/KKM Lemma on the $(k - 1)$-dimensional cuboid to arrive at a contradiction. The cuboid $\mathcal{C}$ is defined as $[0, 1] \times [0, 1 - \epsilon] \times [0, 1 - 2\epsilon] \times \cdots \times [0, (1 - (k - 2)\epsilon)]$. For each point $p = \langle \alpha_1, \cdots \alpha_{k-1} \rangle \in \mathcal{C}$, associate an instance $I(p) = \langle \alpha_1, \alpha_2, \cdots, \alpha_{k-1}, \frac{1}{2} \rangle$ of the $(k, \epsilon, \delta)$-BESTARM. We assign a color from $[k]$ to each point of the cuboid $\mathcal{C}$ as follows: Color of a point $p$ is the arm whose output probability is the largest when the algorithm $A$ is run on input $I(p)$ (breaking the ties arbitrarily). More formally, $color(p) = \arg\max_{a \in [k]} \{\Pr[A(I(p)) = a]\}$. We first claim that the above coloring is a valid Sperner/KKM coloring.

**Coloring of the vertices of $\mathcal{C}$.** A vertex (corner) $v$ of $\mathcal{C}$ is specified by $\langle \alpha_1, \alpha_2, \ldots, \alpha_{k-1} \rangle$ where $\alpha_i$ is either 0 or $(1 - (i - 1)\epsilon)$. Let $i^*$ be the *first index* $i$ where $\alpha_i$ is non-zero if such an $i$ exists otherwise (when all $\alpha_i's$ are 0) $i^* = k$. Note that the $color(v) = i^*$. This is because for any $i \neq i^*$, $a_{i^*} - a_i \geq \epsilon$ and $\mathcal{A}$ should output the arm $i^*$ with probability $\geq 1 - \delta$. Note that all $k$ colors are used while coloring the vertices of $\mathcal{C}$. Figure 2 for the case $k = 3$.

**Coloring of any face $F$.** For an $\ell$ $(0 < \ell < k - 2)$, consider a $(k - 1 - \ell)$ dimensional face $F$ of $\mathcal{C}$. Face $F$ is specified specifying $\ell$ coordinates $i_1 < i_2 < \cdots < i_\ell$ and fixing them to be $\alpha_{i_j}$, where $\alpha_{i_j}$ is either 0 or $(1 - (i_j - 1)\epsilon)$. We claim that the color of any point on $F$ is one of the colors of $\mathcal{V}_F$.

**Case 1.** All $\alpha_i$s are 0 in $F$: In this case the $color(\mathcal{V}_F) = [k] \setminus \{i_1, \ldots, i_l\}$. This is because of the following. Fix $i \in [k] \setminus \{i_1, \cdots, i_\ell\}$. Consider a point $p$ whose $i$th co-ordinate is $(1 - (i - 1)\epsilon)$ and all other co-ordinates are zero. This point belongs to $\mathcal{V}_F$ and gets color $i$. Note that the $k$th-coordinate of $I(p)$ is $1/2$ for every $p$. Thus the color of a point $p$ can not belong to $\{i_1, i_2, \cdots, i_\ell\}$. By a very similar reasoning, it follows that for any point $p$ on $F$ cannot have color from $\{i_1, \ldots, i_l\}$ and hence the coloring is proper.

**Case 2.** At least one $\alpha_{i_j}$ is non-zero: Let $j^*$ be the smallest index such that among $\{1, \cdots, \ell\}$ such $\alpha_{i_{j^*}} \neq 0$. In this case, we claim that the $color(\mathcal{V}_F) = \{1, \cdots i_{j^*}\} \setminus \{i_1, \cdots, i_{j^*-1}\}$. Fix a vertex $v \in \mathcal{V}(F)$. Color of this vertex cannot be any of $i_1, \cdots, i_{j^*-1}$ as all these co-ordinates in $I(v)$ are zero. Let $i > i_{j^*}$, note that $\alpha_{i_{j^*}} - \alpha_i \geq \epsilon$. Thus $i$ can not be the color of $v$. We can apply a very similar reasoning to conclude that for any point $p$ on $F$, its color belongs to $\{1, \cdots i_{j^*}\} \setminus \{i_1, \cdots, i_{j^*-1}\}$.

To finish the proof, we need the following *distance lemma* that can be proved by Data Processing inequality.

**Lemma A.6** (Distance Lemma). *Let $p$ and $q$ be the two points of the cuboid $\mathcal{C}$ such that $\|p - q\|_\infty \leq \nu$. Then, for any $a \in [k]$, $|\Pr[A(I(p)) = a] - \Pr[A(I(q)) = a]| \leq n \cdot k \cdot \nu$*

Now we are ready to complete the proof. Let $\nu = \frac{1}{k^5 n}$. By the Sperner/KKM Lemma, there is $\nu$-neighborhood in $\mathcal{C}$ (in $\ell_\infty$-norm) with points $p_1, \cdots p_k$ such that each point gets a distinct color. Let $a_i = \arg\max_{a \in [k]}\{\Pr[\mathcal{A}(I(p_i)) = a]\}$. Note that all $a_i$'s are distinct as colors of $p_i$'s are distinct. By definition of list replicability $\Pr[A(I(p_i)) = a_i] \geq \frac{1-\delta}{k-1} \geq \frac{k}{(k-1)(k+1)} = \frac{k}{k^2-1}$, $1 \leq i \leq k$. We have the following $\Pr[A(I(p_1)) = a_1] \geq \frac{k}{k^2-1}$ and for $2 \leq i \leq k$, $\Pr[A(I(p_1)) = a_i] \geq \frac{k}{k^2-1} - n \cdot k \cdot \nu = \frac{k}{k^2-1} - \frac{1}{k^4}$ by distance lemma.

Thus, the following leads to a contradiction, $\Pr[A(I(p_1)) \in \{a_1, \cdots a_k\}] \geq \frac{k}{k^2-1} + \frac{k(k-1)}{k^2-1} - \frac{k-1}{k^4} > 1$. $\qquad\square$

### A.5 PROOFS OF SECTION 7

*Proof of Corollary 7.3.* Let $\pi^{\text{init}}$ be a 1-G-optimal design whose existence is guaranteed by Theorem 7.2. For each $a \in \text{supp}(\pi^{\text{init}})$, we add $\lceil d\pi^{\text{init}}(a) \rceil$ copies of $a$ into the support of $\pi$. Note that this implies $\text{supp}(\pi)$ is a multiset. We define $\pi$ to be the uniform distribution over all actions on its support.

Clearly,

$$|\text{supp}(\pi)| \leq \sum_{a \in \text{supp}(\pi^{\text{init}})} \lceil d\pi^{\text{init}}(a) \rceil \leq |\text{supp}(\pi^{\text{init}})| + d \sum_{a \in \text{supp}(\pi^{\text{init}})} \pi^{\text{init}}(a) = 2d.$$

Moreover,

$$V(\pi) = \frac{1}{|\text{supp}(\pi)|} \sum_{a \in \text{supp}(\pi)} aa^\top \succeq \frac{1}{2d} \sum_{a \in \text{supp}(\pi)} aa^\top \succeq \frac{1}{2} \sum_{a' \in \text{supp}(\pi^{\text{init}})} \pi^{\text{init}}(a')aa^\top = \frac{1}{2}V(\pi^{\text{init}}).$$

Therefore, $\pi$ is a $\sqrt{2}$-G-optimal design. $\qquad\square$

*Proof of Theorem 7.4.* We first show correctness. For each batch $b$, let $\mu = (\mu_a)_{a \in \text{supp}(\pi_b)}$. By Hoeffding bound and a union bound, for each batch $b$, with probability at least $1 - \delta/B$,

$$\forall a \in \text{supp}(\pi_b): \quad |\mu'_{a,(b)} - \mu_a| \leq O\left(\frac{\epsilon_b}{\sqrt{d^3}}\right)$$

where $\mu'$ is the empirical mean and $\epsilon_b$ is error parameter defined in Algorithm 4. Thus, by Theorem 5.10, we have that with probability at least $1 - \delta/B$, the estimated mean $\bar\mu_{(b)}$ satisfies $(|\text{supp}(\pi_b)| + 1)$-list replicablity and

$$\forall a \in \text{supp}(\pi_b): \quad |\bar\mu_{a,(b)} - \mu_a| \leq O\left(\frac{\epsilon_b}{\sqrt{d}}\right)$$

Let $V = \sum_{a \in \text{supp}(\pi_b)} aa^\top$. We have

$$\begin{aligned}
\forall a \in \mathcal{A}_b: \langle a, (\theta^* - \hat\theta)\rangle &\leq \langle a(V^\dagger)^{1/2}, V^{1/2}(\theta - \hat\theta)\rangle \\
&\leq \sqrt{a^\top V^\dagger a} \cdot \sqrt{(\theta^* - \hat\theta)^\top V (\theta^* - \hat\theta)} \\
&\leq \sqrt{g(\pi_b)/|\text{supp}(\pi_b)|}\sqrt{(\theta^* - \hat\theta)^\top \Phi^\top \Phi (\theta^* - \hat\theta)} \\
&\leq O(\epsilon_b)
\end{aligned}$$

where we use
$$\|\Phi(\theta^* - \hat\theta)\|_2 = \|\bar\mu_{a,(b)} - \mu_a\|_2 \leq O(\epsilon_b).$$

Hence, the overall regret of the algorithm is with probability at least $1 - \delta$ bounded by,

$$\begin{aligned}
R_T &\leq T_1 \cdot 1 + \sum_{b=2}^{B} T_b \cdot \epsilon_b = T^\alpha + \sum_{b=2}^{B} T_b \cdot \sqrt{\frac{d^4}{T_{b-1}} \log \frac{dB}{\delta}} = T^\alpha + \sqrt{d^4 \log \frac{dB}{\delta}} \sum_{b=2}^{B} \frac{T_b}{\sqrt{T_{b-1}}} \\
&= T^\alpha + \sqrt{d^4 \log \frac{dB}{\delta}} \sum_{b=2}^{B} T^\alpha = T^\alpha + BT^\alpha \sqrt{d^4 \log \frac{dB}{\delta}} \\
&= \tilde{O}\left(d^2 T^\nu \sqrt{T}\right).
\end{aligned}$$

where we ignore constant factors and $\alpha = \frac{1}{2} + \nu$. We also note that $T_B \le T$ and $\sum_{i=1}^{B} T_b = T$

Thus, with probability at least $1 - \delta$, the list complexity is bounded by $(2d+1)^{B-1}$. $\qquad\square$

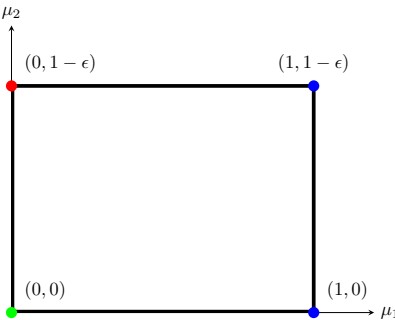

Figure 2: Coloring of 2-dimensional cuboid for $k = 3$. Colors blue, red, and green are identified with coloring with arms 1,2, and 3 respectively. Point $(x, y)$ is associated with instance $(x, y, 1/2)$.

