# OpenReview forum: "Regret-Optimal List Replicable Bandit Learning: Matching Upper and Lower Bounds"
_ICLR.cc/2025/Conference — ICLR 2025 Poster_

### Official Review · Reviewer_m3DX · 2024-10-30

**Soundness:** 4
**Presentation:** 4
**Contribution:** 3
**Rating:** 8
**Confidence:** 4

**Summary:**

This paper introduces the concept of list replicability to the multi-armed bandit model, where the sequence of arm pulls must lie in a small finite list with high probability. The authors design and analyze three algorithms, each providing different levels of guarantees on list replicability and high-probability regret. Additionally, a nearly matching lower bound is proved for any algorithm with sub-linear regret. The paper also extends the study to linear bandits.

**Strengths:**

1. Although the paper is highly theoretical, it is well-presented and clearly conveys the key ideas behind the algorithm designs and proofs.

2. Three algorithms with varying levels of guarantees are introduced, each with its own significance. Notably, the first algorithm achieves near-optimal cumulative regret, and the total number of possible traces is independent of T. The last algorithm is based on a subroutine from Dixon et al. (2023) and is nearly optimal, given the lower bound in Section 6.

3. The theoretical contributions are nontrivial, and the analysis of the phase-elimination algorithm is novel, which should be of interest to the bandit community. It is also interesting that the lower bound is proven using the Sperner/KKM lemma, a combinatorial result in coloring.

**Weaknesses:**

The main criticism of the paper might lie in its motivation. In the introduction, it is suggested that list replicability might be beneficial for safety-critical applications, as one could be prepared for the action sequence being played. However, although the proposed algorithms can ensure a small number of traces with high probability, these possible traces cannot be known without exact knowledge of the problem instance. Therefore, outside of the theoretical domain, the practical application of list replicability seems limited.

**Questions:**

1. Could you compare $\rho$-replicability and list replicability with respect to their potential practical applications, such as in clinical trials?
2. Why is $C$ referred to as the number of shifts? Do you mean the number of possible shift $r$?
3. Minor typos: Line 207: Theorem 2.1 -> Assumption 2.1; Line 210: lemma -> lemmas; Line 346: the of -> the number of.
4. Thomson sampling and UCB are two well-established algorithms in the bandit literature. Thomson sampling is randomized, making it tricky to provide strong list replicability guarantees. Could you discuss the potential challenges in adapting UCB? My intuition is that UCB might achieve good list replicability with appropriate early-stage modifications.

---

> ### Author Response · Authors · 2024-11-22
>
> Thank you for the review.
>
> Response to Weaknesses
>
> Formalizing replicability is an emerging research direction and this paper introduces and explores one viable notion. The practical applications of all proposed replicability notions (including ours) need to be investigated thoroughly in collaboration with domain experts.
>
> Response to Questions
> 1. $\rho$-replicability requires a *good seed* to be shared among the independent runs to achieve replicability. However, there is no pragmatic way of checking if a seed is good or not. List replicability does not have such a restriction. So long as samples are being drawn for the same distribution list replicability is guaranteed.
> 2. $C$ is the total number of shifts, $r$ is uniformly sampled from $[C]$. We have $C=12k/\delta$ and setting $\delta=\frac{1}{2}$ (say) yields in $C=24k$ total possible shifts.
> 3. Thank you for pointing out the typos. They have been corrected in the updated version.
> 4. While UCB algorithms have several desirable properties, they seem to be ill-suited to ensure replicability. In particular, the list complexity of UCB-based algorithms seems to be exponentially dependent on $T$. Consider the case when $k=2$ and both arms have identical distributions. Let $UCB_1$ and $UCB_2$ represent the UCB estimates of arm 1 and arm 2, respectively. Let us assume that at some time $t$, $UCB_1<UCB_2$. UCB algorithm dictates that we shall play arm 2 till the estimate of arm 1 is larger. However, because the samples are random, the time at which we start playing arm 1 is probabilistic. For each time step where a switch is possible results in a new trace. This trait makes the list complexity of UCB-based algorithms exponentially dependent on $T$. Note that the list complexities of our algorithms are independent of $T$. This discussion has been added to the updated version. It is unclear how an early modification could yield low list replicability when $T$ is large and when distributions are similar.

---

> > ### Comment · Reviewer_m3DX · 2024-11-25
> >
> > Thank you for your response. I will maintain my original rating.

---

### Official Review · Reviewer_jzZs · 2024-11-01

**Soundness:** 3
**Presentation:** 3
**Contribution:** 3
**Rating:** 8
**Confidence:** 3

**Summary:**

The paper introduces replicability to the multi-armed bandit area through the concept of list replicability and proposes algorithms for both k-armed and linear bandits. Notably, for k-armed bandits, the authors provide a lower bound demonstrating that one proposed algorithm is nearly optimal.

**Strengths:**

The paper is well-written and structured, with a clear motivation. Tho short, it presents a comprehensive set of results for both k-armed and linear bandits, though the linear bandit results appear to be preliminary.

**Weaknesses:**

- It would be helpful to clarify which variables the hidden logarithmic factors depend on, and whether these factors are consistent throughout the paper.
- No experiments are presented.

**Questions:**

- While it seems that replicability papers often omit experiments, bandit experiments are generally straightforward to conduct. Did the authors consider demonstrating some experimental results?
- Most of the algorithms appear to be adaptations of standard elimination-based bandit algorithms for both k-armed and linear bandit problems. It would be valuable if the authors could reference each classical elimination algorithm and include a side-by-side comparison showing what aspects of these algorithms break replicability and how the new modifications enable it.
- Given that the study addresses regret minimization—typically dominated by UCB-type algorithms for stronger instance guarantees—the authors’ choice of elimination-based algorithms is interesting. Could you clarify the rationale behind this choice?

---

> ### Author Response · Authors · 2024-11-22
>
> Thank you for the review.
>
> Addressing Weaknesses:
> 1. Thank you for the suggestion. $\tilde{O}(\cdot)$ hides $\log k$, $\log\log T$, $\log 1/\delta$ factors. This has been clarified in the updated version. These factors are consistent throughout the entire paper.
> 2. This work focuses on theoretical foundations, the main contribution is to define and introduce the notion of list replicability in MAB and explore possibilities and impossibilities.
>
> Addressing Questions:
> 1. The focus of this work is to theoretically formalize list replicability in the context of MAB and design list replicable algorithms and prove impossibility results.
> 2. Known phase elimination algorithms work by estimation algorithms run in batches, and during each batch they estimate the means of arms and eliminate the arms based on certain criteria. These algorithms as presented have a list size around $O(\ell^B)$, where $\ell$ depends on the list complexity of the estimators, this itself could be very large. So new algorithms and analysis are needed.
> In Algorithm 1, the modification is the choice of hyperparameters and a novel analysis. In algorithm 2, the main modification is the introduction of random shifts which brings new challenges in the analysis. The third algorithm uses list-replicable estimators. We will add this discussion as a remark.
>
> 3. While UCB algorithms have several desirable properties, they seem to be ill-suited to ensure replicability. In particular, the list complexity of UCB-based algorithms seems to be exponentially dependent on $T$. Consider the case when $k=2$ and both arms have identical distributions. Let $UCB_1$ and $UCB_2$ represent the UCB estimates of arm 1 and arm 2, respectively. Let us assume that at some time t, $UCB_1<UCB_2$. UCB algorithm dictates that we shall play arm 2 till the estimate of arm 1 is larger. However, because the samples are random, the time at which we start playing arm 1 is probabilistic. For each time step where a switch is possible results in a new trace. This trait makes the list complexity of UCB-based algorithms exponentially dependent on $T$. Note that the list complexities of our algorithms are independent of $T$. This discussion has been added to the updated version.

---

> > ### Comment · Reviewer_jzZs · 2024-11-23
> >
> > Thank you for the response. I have increased my rating.

---

### Official Review · Reviewer_ksHy · 2024-11-03

**Soundness:** 3
**Presentation:** 3
**Contribution:** 3
**Rating:** 6
**Confidence:** 3

**Summary:**

This paper studies list replicability in multi-armed bandits and linear bandits. It comes up with the notion of $(\ell, \delta)$-list replicability, and proved various trade-off between replicability and regret dependency on number of arms and on time horizon. Furthermore, the paper extends the results to linear bandits setting.

**Strengths:**

1. The paper proposes a definition of reproducibility in bandits problems.
2. The paper proves tight trade-off between replicability and regret dependency on $T$.
3. The proof to the lower bound is quite insightful.

**Weaknesses:**

1. The algorithms are generally based on successive elimination, so it contains less insight on more widely used bandits algorithms like UCB.
2. The proofs to the upper bounds are quite simple and lack enough novelty given their similarity to successive elimination.

**Questions:**

1. Line 18, $\widetilde O \sqrt{kT}$ missing parentheses.
2. The notion of $O(\cdot)$ and $\Omega(\cdot)$ was a little abused. The paper contains regret bound like $\widetilde O (k^{\frac32} T^{\frac12 + 2^{-\Omega(B)}})$. Here, it's inappropriate to use $\Omega(\cdot)$ in $\widetilde O(T^{2^{-\Omega(B)}})$, because the constant before $B$ cannot be ignored, e.g., $T^{2^{-B}}$ and $T^{2^{-2B}}$ have very different order.

---

> ### Author Response · Authors · 2024-11-22
>
> Thank you for the review.
>
> Response to Weaknesses:
> 1. For the purposes of list replicability, phase elimination algorithms have the desirable property that if hyperparameters are picked correctly then the arms are deleted in one of two consecutive rounds. However, the list complexity of UCB-based algorithms is exponentially dependent on $T$. Consider the case when $k=2$ and both arms have identical distributions. Let $UCB_1$ and $UCB_2$ represent the UCB estimates of arm 1 and arm 2, respectively. Let us assume that at some time $t$, $UCB_1<UCB_2$. UCB algorithm dictates that we shall play arm 2 till the estimate of arm 1 is larger. However, because the samples are random, the time at which we start playing arm 1 is probabilistic. For each time step where a switch is possible results in a new trace. This trait makes the list complexity of UCB-based algorithms exponentially dependent on $T$. Note that the list complexities of our algorithms are independent of $T$.
>
> 2. The first algorithm achieves optimal regret, and we bound the list size in a way that is independent of $T$. Similarly, the second algorithm achieves optimal regrets when $k$ and $\delta$ are constant.
> UCB algorithm, though achieving optimal regret, has high list complexity. We do not know if the UCB algorithm can be modified to get a small list size while achieving optimal regret. The base phase elimination algorithm itself does not have $2^k$ list replicability. Only when the parameters are correctly chosen do we get a list size of $2^k$. As pointed out by the reviewer m3DX, our theoretical contribution is a novel analysis of the phase elimination algorithm (aided by careful choice of parameters) to exhibit a list size of $2^k$. We then show how we can reduce this to $O(k/\delta)$ by applying a novel technique of shifting the deletion criteria. All contributions are theoretical in nature and are novel as they help set foundations for studying list replicability in more general reinforcement learning settings.
>
> Response to Questions:
> 1. Thank you for the correction. It has been corrected in the updated version.
> 2. There is no attached constant multiplicative constant to $B$. So regret is simply $\tilde{O}(k^{3/2}T^{1/2+2^{-(B+1)}})$.  The updated version fixes this.

---

> > ### Comment · Reviewer_ksHy · 2024-11-30
> >
> > Thanks for the response. I'll update my review.

---

### Official Review · Reviewer_ujbd · 2024-11-04

**Soundness:** 3
**Presentation:** 3
**Contribution:** 2
**Rating:** 3
**Confidence:** 4

**Summary:**

This paper studies list replicability in multi-armed bandits (MAB), defining an algorithm as list replicable if it limits the distinct arm sequences (traces) across independent executions with high probability. Further, this paper proposes three algorithms with different parameters of list replicability. Finally, this paper investigates a lower bound of bandits with list replicability.

**---After rebuttal---**

My primary concern pertains to the main claims of the paper, as highlighted in the title: "Regret-Optimal" and "Matching Upper and Lower Bounds." Following a detailed discussion with the authors, I have observed that the paper fails to provide any lower bound in terms of regret for their setting, even in Section 6. Consequently, the claims of "Regret-Optimal" and "Matching Upper and Lower Bounds" appear highly questionable.

In their latest response, the authors stated that their claim of "regret-optimality" is based on achieving a $\tilde{O}(\sqrt{T})$ regret. However, to the best of my knowledge in the field of bandits, it is not standard practice to assert optimality of regret solely with respect to the parameter $T$, while disregarding other critical parameters (e.g., $K$). Given this significant issue of overclaim, I am unable to recommend this paper for acceptance.

**Strengths:**

1. The problem setting proposed is both novel and intriguing, characterized by a rigorously defined concept of bandit replicability in Definition 2.2.
2. The theoretical analysis provided is exhaustive, introducing three distinct algorithms tailored to various parameters of replicability.

**Weaknesses:**

1. Algorithms 1 and 2 exhibit considerable similarities. Could there be a method to consolidate these two algorithms into a unified framework?

2. In Theorem 6.1, the designation "lower bound" appears misapplied as it does not seem to correspond to the lower bounds of any algorithms discussed previously. Notably, in Theorem 6.1 we have $l \approx k$, whereas in prior algorithms $l \gg k$ in most cases. In my humble opinion, a valid lower bound should be able to explain whether the proposed algorithms can be further optimized in general.
Furthermore, why the authors said "We show that result (3) is nearly tight for B=2" in the abstract. What's the hidden constant behind $\Omega(B) $ in (3). Do you mean the regret of (3) is $O(T)$ for $B=2$?

3. Would it be more accurate to describe what is currently referred to as "lower bounds" in Theorem 6.1 as "impossibility results"? I think Theorem 6.1 is quite trivial because any pair of traces should share more than two arms if the total number of traces is less than $K$.

4. The absence of experimental validation in this paper is notable. Including even preliminary numerical simulations or toy experiments could significantly enhance the validity and impact of the proposed algorithms.

**Questions:**

See weaknesses.

---

> ### Author Response · Authors · 2024-11-22
>
> Thank you for the review.
>
> Addressing Weaknesses and Questions:
> 1. The main difference between Algorithms 1 and 2 is that Algorithm 2 uses a random threshold to eliminate the arms and thus needs a better estimate compared to Algorithm 1. To do so, the batch length needs to be changed appropriately. As pointed out, Algorithms 1 and 2 can be unified, however, we believe that presenting them separately helps with the clarity of the analysis.
>
> 2. Algorithm 1 gives a list size of $2^k$, Algorithm 2 gives a list size of $O(k/\delta)$, and Algorithm 3 gives a list size of $k$ (with $B = 2$). A natural question is whether the list sizes can be further reduced. Our lower bound result (Theorem 6.1) states that the list size cannot be less than $k$, if the regret were to be sub-linear.  We can restate the theorem as follows: "Any algorithm with $o(T)$ regret must have a list complexity $\geq k$". On the other hand, we know there is a $o(T)$-regret algorithm with list complexity equals $k$ (Third algorithm with $B = 2$).  On comment about,  $\Omega(B)$, this can be changed to $2^{-(B+1)}$ so that there are no hidden constants. This has been updated in the latest version. When the regret is less $o(T^{2/3})$, our algorithms have a list complexity asymptotically larger than $k$.  It is an open question whether we can design algorithms with smaller list complexity. This is stated as an open question in the conclusions.
>
> 3. We do not see how "sharing an arm across two different traces" yields a lower bound on the list complexity. In fact, any reasonable algorithm must play all the arms with a very high probability; otherwise, the missing arm could have the highest mean and suffer a regret of $O(T)$. So in any algorithm with sublinear regret all arms are shared across (almost) all traces.  We do not see a trivial way of arguing the lower bound. The fact we are looking for algorithms with sublinear regret is crucial as it is trivial to design algorithms with linear regret with just one trace.
>
> 4. Thank you for the suggestion. This work focuses on the theoretical foundations, the main contribution is to define and introduce the notion of list replicability in MAB and explore possibilities and impossibilities.

---

> > ### Comment · Reviewer_ujbd · 2024-11-25
> >
> > Thank you for your detailed response. Points 1 and 3 are now clear.
> >
> > Regarding point 2: I understand the statement that "any algorithm with regret o(T)
> > must have a list complexity >=k.” However, your claim that "it almost exactly matches the k-list replicable upper bound for B=2" is somewhat unclear and potentially misleading. If I understand correctly, the tightness of your result pertains primarily to the value of the first parameter of your (xx,xx)-list replicability in some cases, rather than to the regret itself. In the context of bandits, when we discuss upper bounds matching lower bounds, we typically refer to regret. Similarly, in Section 6 of your paper, there does not appear to be a lower bound in terms of regret.
> >
> > Regarding point 4: While I recognize that the primary contribution of your work lies in its theoretical contribution, it is crucial to verify its availability even with a toy simulation,  as many other theoretical papers did.

---

> ### Author Response · Authors · 2024-11-26
>
> Thanks for the suggestion. We made changes in the abstract to accommodate your suggestion regarding point 2. We hope it is more clear and less confusing now. Accordingly, we have updated the remark in lines 453-459.
>
> Regarding experimentation in point 4, we have conducted preliminary experimentation. We compared Algorithm 1 with the UCB algorithm. with $k=2,3,4,5$ arms. We performed 100 independent runs with a horizon of $T=10^6$, and the distributions of the $k$ arms were chosen to be normal distributions with means equispaced between $[0,1]$ each with standard deviation 1. For each algorithm and  $k$, we keep track of the unique traces encountered over the course of the 100 independent runs. We adjusted the hyperparameters so that UCB and Algorithm 1 have similar regrets ($c=0.04$ for Alg 1 and $c=1$ for UCB). Then, we compare the list complexities. Here are the results:
>
> |Algorithm Type		|	Number of Arms ($k$)|		Number of traces observered 		|	Avg. Expected Regret($\times10^{-4}$)	|
> |-------------------|:-----------------:|:-----------------------------------------:|:-----------------------------------------:|
> |Algorthim 1	    |         2         |       			2   					| 					0.11					|
> |Algorthim 1	    |         3         |       			3   					| 					0.99					|
> |Algorthim 1	    |         4         |       			4   					| 					1.40					|
> |Algorthim 1	    |         5         |       			5   					| 					1.67					|
> |UCB			    |         2         |       			100   					| 					0.27					|
> |UCB			    |         3         |       			100   					| 					0.61					|
> |UCB			    |         4         |       			100   					| 					1.00					|
> |UCB			    |         5         |       			100   					| 					1.41					|
>
> UCB encounters a new trace every single run. Whereas Algorithm 1 seems to encounter traces proportional to $k$ when the distributions are equispaced between $[0,1]$.

---

> > ### Comment · Reviewer_ujbd · 2024-11-26
> >
> > **Regarding the experiment**: I appreciate the authors' efforts in conducting the experiment. Indeed, this is more of a "preliminary" experiment. For instance, using UCB as a baseline does not seem like a good choice since it's nearly non-replicable. Additionally, the results for $K=2$ are puzzling, suggesting that UCB performs even worse, which needs further explanation. Despite these issues, based on what the authors have done so far, I believe they can deliver a complete version of their simulation in the final paper. I am confident about it.
> >
> > **Regarding the claim in their abstract**: I am not persuaded that merely adding lines 453-459 and changing a few sentences in the abstract make it "more clear and less confusing." As I previously mentioned, there's no actual lower bound in terms of regret. Therefore, any claims about the tightness of regret should be revised. This is particularly crucial in the title. I believe the claim of "regret-optimal" is not true.
> >
> > Moreover, if the primary claim, as stated in the title "Regret-Optimal List Replicable Bandit Learning: Matching Upper and Lower Bounds," proves to be inaccurate, does this paper necessitate a new round of review for the next conference? I have a strong concern about it.
> >
> > I am decreasing the score to 3, mainly based on what I said in "regarding the claim in their abstract" above. I remain open to adjusting the score as necessary.

---

> > > ### Author Response · Authors · 2024-11-26
> > >
> > > Thank you for your feedback and concerns. We respectfully disagree with your assessment regarding the regret bounds. The algorithms (1 and 2) proposed in our paper are nearly regret-optimal $\tilde{O}(\sqrt{T})$, as they share the same regret bound wrt $T$, which have been shown in the literature to be near-optimal. The change we made in the abstract clearly explains a contribution: sub-linear regret algorithms must have a list complexity of at least $k$. If the phrase "regret-optimal" in the title or elsewhere is misleading, we are open to suggestions for acceptable wording. We would like to point out that the statements of the theorems in the paper are precise and are not misleading.

---

### Meta-Review · Area_Chair_pkeZ · 2024-12-23

**Metareview:**

This is a borderline paper on a current topic of interest: replicability of experiments considered within the bandit context. Some of the criticisms given by the reviewers concerns the claim of matching upper and lower bounds, (the matching parts refers only to the dependence on T), and that classical algorithms like UCB are not covered by the replicability definition.  Despite these criticisms the paper provides new ideas that might turn out to be valuable to the bandit community.

**Additional Comments On Reviewer Discussion:**

There was a significant exchange between reviewers and authors. It did not alter the scores however.

---

### Decision · Program_Chairs · 2025-01-22

Accept (Poster)